# Genome-Wide Identification and Comprehensive Analysis of the GARP Transcription Factor Superfamily in *Populus deltoides*

**DOI:** 10.3390/genes16030322

**Published:** 2025-03-09

**Authors:** Qin Yang, Zhengquan He, Chenjia Zheng, Ruoyu He, Yu Chen, Renying Zhuo, Wenmin Qiu

**Affiliations:** 1Key Laboratory of Three Gorges Regional Plant Genetic & Germplasm Enhancement (CTGU)/Biotechnology Research Center, China Three Gorges University, Yichang 443002, China; 13058636396@163.com (Q.Y.); zhq_he@163.com (Z.H.); 18272122310@163.com (R.H.); 2State Key Laboratory of Tree Genetics and Breeding, Key Laboratory of Tree Breeding of Zhejiang Province, Research Institute of Subtropical Forestry, Chinese Academy of Forestry, Hangzhou 311400, China; 20221242@mails.jlau.edu.cn (C.Z.); zhuory@gmail.com (R.Z.); 3School of Horticulture, Jilin Agricultural University, Changchun 130118, China; 4Agricultural Technology Extension Centre of Dongtai, Yancheng 224200, China; chenyu199601201@163.com

**Keywords:** *Populus deltoids*, GARP transcription factor superfamily, phosphorus and nitrogen deficiency, chlorophyll content

## Abstract

**Background/Objectives:** The GARP transcription factor superfamily is crucial for plant growth, development, and stress responses. This study systematically identified and analyzed the *GARP* family genes in *Populus deltoides* to explore their roles in plant development and abiotic stress responses. **Methods:** A total of 58 *PdGARP* genes were identified using bioinformatics tools. Their physicochemical properties, genomic locations, conserved motifs, gene structures, and phylogenetic relationships were analyzed. Expression patterns under phosphorus and nitrogen deficiency, as well as tissue-specific expression, were investigated using RT-qPCR. Transgenic RNAi lines were generated to validate the function of *GLK* genes in chlorophyll biosynthesis. **Results:** The 58 *PdGARP* genes were classified into five subfamilies based on their evolutionary relationships and protein sequence similarity. Segmental duplication was found to be the primary driver of the PdGARP family’s expansion. *Cis*-regulatory elements (CREs) related to light, hormones, and abiotic stresses were identified in the promoters of *PdGARP* genes. Differential expression patterns were observed for NIGT1/HRS1/HHO and PHR/PHL subfamily members under phosphorus and nitrogen deficiency, indicating their involvement in stress responses. KAN subfamily members exhibited tissue-specific expression, particularly in leaves. Structural analysis of the GLK subfamily revealed conserved α-helices, extended chains, and irregular coils. Transgenic RNAi lines targeting *GLK* genes showed significant reductions in chlorophyll and carotenoid content. **Conclusions:** This study provides a comprehensive analysis of the GARP transcription factor superfamily in *P. deltoides*, highlighting their potential roles in nutrient signaling and stress response pathways. The findings lay the foundation for further functional studies of *PdGARP* genes and their application in stress-resistant breeding of poplar.

## 1. Introduction

The GARP transcription factor superfamily, related to the MYB superfamily and named after maize GOLDEN2, ARR B-class, and *Chlamydomonas* Psrl [1] is characterized by a myb-like DNA-binding domain found exclusively in plant proteins [2]. This domain, featuring the conserved motif “SHAQKYF”, is crucial for recognizing and binding specific DNA sequences [3]. Initially identified in *Arabidopsis thaliana*, *GARP* genes have since been found in various plant species, including in *Spirodela polyrhiza*, *Medicago truncatula*, *Camellia sinensis*, *Malus domestica*, and *Brassica napus* [4,5,6,7,8]. These genes play vital roles in hormonal signaling, nutrient response, chloroplast biogenesis, and plant development [9,10,11,12,13]. The GARP superfamily is divided into five subfamilies: ARR-B, GLK, NIGT1/HRS1/HHO, KAN, and PHR/PHL. Among them, ARR-Bs are key to cytokinin signaling [14,15], while *KAN* genes control leaf polarity [16,17,18,19,20]. *NIGT1/HRS1/HHO* genes regulate nitrate-inducible genes [21,22,23,24,25] and *PHR/PHL* are involved in phosphorus deficiency response [26,27,28]. *GLK* genes, known for their conserved function, regulate chloroplast development and chlorophyll biosynthesis [29,30,31,32,33,34,35,36].

*Populus deltoides*, commonly known as poplar, is a fast-growing tree species introduced to China. It has strong adaptability and a high output rate [37]. However, its output rate has been affected by many factors. Under low nitrogen stress, poplar growth and chlorophyll synthesis are inhibited, which ultimately reduces output rate. The plant height and biomass of poplar are significantly inhibited under phosphorus deficiency. This indicates that nitrogen and phosphorus deficiency can seriously affect the biomass production of poplar [38]. For green plants, chloroplast development is also essential for plant growth [39]. It not only determines the efficiency of photosynthesis but also plays key roles in many physiological processes, such as phytohormones synthesis, stress tolerance, and energy production. Phosphorus deficiency, nitrogen deficiency, and chloroplast development have important effects on the growth of poplar. The GARP transcription factor superfamily plays an important role in response to phosphorus deficiency, nitrogen deficiency, and chloroplast development [40]. It is necessary to conduct a comprehensive analysis of the GARP transcription factor superfamily in *P. deltoides*.

In the present study, 58 *PdGRAP* genes were identified in *P. deltoides* and their motif, chromosomal arrangement, and gene structure were analyzed. The expression patterns of the PHR/PHL and NIGT1/HRS1/HHO subfamilies in response to phosphorus and nitrogen deficiency stresses were also analyzed. The heatmap shows the expression patterns of *KAN* gene members in different tissues of *P. deltoides*. In addition, *RNAi-GLK* lines were obtained, and chlorophyll content was determined. This study will enhance our understanding of the molecular characteristics of the PdGARPs and further facilitate functional studies of *PdGARPs* in *P. deltoides*.

## 2. Materials and Methods

### 2.1. Identification of GARP Family in Plants

In this study, the genomic and protein sequences from *A. thaliana* and *P. deltoides* were downloaded from the Phytozome Genome Database (https://phytozome-next.jgi.doe.gov/ (accessed on 5 August 2024)). To identify the *GARP* genes in *P. deltoides*, 56 GARP protein sequences from Arabidopsis were used to perform a BLAST alignment against the poplar protein database by TBtools-II software (v2.149) [41]. Sequences with a similarity of more than 36% and E-values less than 8 × 10^−62^ were retained as candidate GARP proteins in poplar. These candidate GARP proteins were designated as *P. deltoides* GARPs (PdGARPs) and were named according to their chromosomal positions. The number of amino acids, molecular weight (MW), isoelectric point (pI), instability index, aliphatic index, grand average of hydropathicity (GRAVY), and other properties of each GARP protein were predicted using the TBtools software.

### 2.2. Phylogenetic Tree Construction of PdGARPs

The conserved motifs of PdGARP proteins were elucidated using the Multiple Em for Motif Elicitation (MEME) program (MEME Suite 5.5.7; https://meme-suite.org/meme/tools/meme (accessed on 19 August 2024)), with the number of motifs set to 10.

TBtools software was employed to extract the exon–intron data for PdGARPs from the coding and genomic sequences of the related genes and visualize the conserved motifs and gene structures.

TBtools was used to extract the promoter region, a 2.0 kb upstream sequence of the start codon, for each *PdGARP* gene and to identify the *cis*-regulatory elements (CREs). The extracted sequences were then submitted to PlantCARE (https://bioinformatics.psb.ugent.be/webtools/plantcare/html/ (accessed on 3 September 2024)). Finally, ChiPlot (https://www.chiplot.online/ (accessed on 12 September 2024)) was used to visualize the results, including the types and numbers of CREs.

### 2.3. Chromosomal Distribution, Gene Duplication, and Synteny Analyses of PdGARPs Genes

According to the genome of *P. deltoides* and its annotation file, gene duplication of *PdGARPs* was analyzed using TBtools. Thereafter, TBtools was employed to determine the nucleotide substitution parameters, including Ka (nonsynonymous) and Ks (synonymous), followed by the calculation of the Ka/Ks ratio. The MCScanX plugin in TBtools was employed for synteny analysis of *P. deltoides*. Finally, the Circos plots of PdGARPs were generated by the Circos plugin in TBtools.

To study the sequence similarities and secondary structure information of the GLK subfamily in *P. deltoides*, sequence alignment was performed using ESPript 3.0 (https://espript.ibcp.fr/ESPript/ESPript/ (accessed on 6 November 2024)). Protein structure homology modeling of the GLK subfamily in *P. deltoides* was then carried out using SWISS-MODEL (https://swissmodel.expasy.org/ (accessed on 24 December 2024)).

Data processing and graphical representation were performed using GraphPad Prism 10 and Excel (Microsoft).

### 2.4. Plant Material and Treatments

Experimental materials, including wild-type (WT) and transgenic poplar lines, were planted from the tree tissue culture laboratory at the Research Institute of Subtropical Forestry, CAF, Zhejiang Province, China. The photoperiod was 16 h light/8 h dark, with a temperature of 25 ± 1 °C and an air humidity of 70%. Robust poplar seedlings with similar growth status were selected for this experiment. The *P. deltoides* seedlings were transplanted into glass tissue culture tubes containing 10 mL of modified Hoagland nutrient solution [42].

To further analyze the expression patterns of *NIGT1/HRS1/HHO* genes under nitrogen deficiency stress, Ca (NO_3_)_2_·4H_2_O was replaced with CaCl_2_, and KNO_3_ was replaced with KCl in the nutrient solution. For the nitrogen deficiency treatment, 12 seedlings were sampled at 0, 1, 4, and 7 days. Similarly, the expression levels of genes were measured in response to phosphorus deficiency stress. KH_2_PO_4_ was replaced with KCl in the nutrient solution. For the phosphorus deficiency treatment, 12 seedlings were sampled at 0, 1, 4, and 7 days. The chemical reagents used to prepare the nutrient solution are all from Sinopharm Chemical Reagent Co., Ltd., Shanghai, China.

The aforementioned whole seedlings were sampled for RNA extraction. Three independent biological replicates were included at each sampling time point to analyze the gene expression levels of *NIGT1/HRS1/HHO* genes and *PHR/PHL* genes under phosphorus and nitrogen deficiency stresses. Total RNA was extracted using the RNAprep Pure Plus Kit (TIANGEN, Beijing, China) and reverse-transcribed into cDNA using the PrimeScript™ II 1st Strand cDNA Synthesis Kit (TAKARA, Beijing, China). RT-qPCR was performed using 2× Q3 SYBR qPCR Master Mix-Universal (TOLOBIO, Shanghai, China). The primers for this study were designed using the Primer3Plus website (https://www.primer3plus.com/ (accessed on 14 August 2024)) (Appendix A). All RT-qPCR experiments use Actin-RT-F/R as the housekeeping genes.

Fragments of the conserved regions of the GLK subfamily were cloned and inserted into the RNAi vector pANDA35HK forward and backward. To obtain RNA interference lines, these RNAi vectors were then introduced into the *P. deltoides* genome via agrobacterium-mediated transformation. Chlorophyll (Chl) and carotenoids (Car) were extracted using 95% ethanol. After grinding and filtering the extract, the absorbance was measured at 470, 645, and 663 nm using a spectrophotometer (Thermo Fisher Scientific, Waltham, MA, USA). The contents of Chla, Chlb, total chlorophyll (Chl (a + b)), and Car were then calculated according to the formula [43].

## 3. Results

### 3.1. Identification of PdGARP Members in P. deltoides

Using BLASTP searches in the Phytozome Genome Database, we identified 58 PdGARP proteins in *P. deltoides*. The genomic locations of the identified *GARP* genes were physically mapped onto the chromosomes of *P. deltoides*. The 58 *PdGRAP* genes were named based on their position on their respective chromosomes. Ultimately, a total of 56 *PdGARPs* were mapped onto 19 chromosomes in *P. deltoides*, and 2 additional *GARPs* were mapped onto scaffold (Figure 1). Clearly, there is only one *GARP* gene on chromosomes Chr04, 05, 09, and 15, whereas Chr08 contained the highest number of *GARP* genes.

The phylogenetic tree of *A. thaliana* and *P. deltoides* was constructed and analyzed to clarify the evolutionary relationships and functional divergence of GARP proteins in *P. deltoides* (Figure 2). PdGARPs were classified into five subfamilies based on their evolutionary relationships and protein sequence similarity, including ARR (15 members), GLK (8 members), NIGT1/HRS1/HHO (6 members), KAN (10 members), and PHR/PHL (20 members). The differences in the number of PdGARPs within these subfamilies indicate a distinct expansion trend among these subfamilies.

An overview of the characterization of PdGRAPs is analyzed in Appendix A. The results showed that the number of amino acids and molecular weight of the 58 PdGARP proteins ranged from 282 amino acids (PdGARP1) to 714 amino acids (PdGARP43) and from 31.69 kDa (PdGARP1) and 78.31 kDa (PdGARP43), respectively. The aliphatic index ranged from 49.72 (PdGARP50) to 84.23 (PdGARP5). The GRAVY value ranged from −1.044 (PdGARP34) to −0.319 (PdGARP33), and the theoretical pI ranged from 5.17 (PdGARP16) to 9.36 (PdGARP11). The instability index ranged from 37.47 (PdGARP11) to 71.34 (PdGARP34).

### 3.2. Conserved Motif and Gene Structure Analyse

To further elucidate the potential function of PdGARP proteins, an in-depth analysis of their domain and conserved motif composition was conducted using the MEME suite. Firstly, a total of 10 putative conserved motifs of 58 PdGARP proteins were identified (Figure 3a). The number and arrangement of motifs were similar within the same subfamily, indicating that the functions of these protein are conserved in certain subfamilies. However, motifs 4 and 8 were type-specific motifs. Specifically, motif 4 was found only in PHR/PHL subfamily, and the KAN subfamily contained the unique motif 8. These results suggest that protein sequences and functions of *PdGARP* genes vary significantly across different subfamilies, reflecting evolutionary divergence among the tested species. Overall, the results indicate substantial variation in both protein sequences and functions among different *PdGARP* gene types, with clear evolutionary differences across the five subfamilies.

To assess the sequence diversity of PdGARPs, the intron/exon structures were detected for each *PdGARP*. Most *PdGARPs* had six exons and six introns, while *PdGARP8* contains 12 introns and exhibited variation in length (Figure 3b,c). This suggests that it may affect transcriptional regulation and alternative splicing of genes. However, in general, the majority of *PdGARPs* in the same subfamilies had similar gene structures.

### 3.3. Gene Duplication and Synteny Analysis of PdGARPs

To explore the evolutionary fate of PdGARPs, the ratio of nonsynonymous (Ka) to synonymous (Ks) substitution for each duplicated *PdGARP* gene pair was calculated (Table 1). The Ka/Ks ratios for all duplicate gene pairs were less than 1, ranging from 0.1934 (*PdGARP29*–*PdGARP31*) to 0.5966 (*PdGARP6*–*PdGARP42*). This indicates that these genes are under purifying selection.

To clarify the conservation of synteny and the relationships of orthologs among the *PdGARP* genes, a collinearity analysis was conducted for replication events within the PdGARP family. We found that 42 genes formed 26 segmental duplication events on chromosomes using the MCScanX (Figure 4). However, tandem duplication occurred only in two pairs of genes. These results suggest that segmental duplication is the primary driver of the evolutionary expansion of the *PdGARPs* gene family in *P. deltoides*.

The evolutionary relationship of *GARP* genes was analyzed by constructing collinearity relationship between *P. deltoides* and *A. thaliana*. The collinearity analysis revealed that there were 50 gene pairs shared between *P. deltoides* and *A. thaliana* (Figure 5). Notably, multiple *PdGARP* genes (such as *PdGARP18*, *PdGARP21*, and *PdGARP51*) were found to be associated with at least two syntenic gene pairs, suggesting that these *GARP* genes may play an important role in the evolution of the *GARP* gene family. The collinearity analysis between *P. deltoides* and *A. thaliana* suggests that there are minor differences in the chromosomal structure of the two species. This indicates that the order in which the *GARP* genes are arranged on chromosomes has not changed much during evolution. It helps us better understand the genetic relationships between species, their evolutionary history, and their adaptation strategies.

### 3.4. Cis-Element Analysis of PdGARPs

To investigate the underlying regulatory mechanisms of *GARP* genes in response to abiotic stresses and hormones, the *cis*-regulatory elements (CREs) in the 2000 bp promoter region upstream of each *GARP* gene were analyzed using the plantCARE database. The results showed that each *PdGARP* gene contains stress- and hormone-related CREs in its promoter region (Figure 6). A total of 1274 CREs were identified across the promoter regions of *PdGARP* genes, including 663 light-responsive CREs, 135 anaerobic-induced CREs, 98 abscisic acid (ABA)-responsive CREs, and 82 methyl jasmonate (MeJA)-responsive CREs. Among these, the most abundant CREs in the GARP gene promoters were light-responsive elements (663), while the least abundant were auxin response elements, elements involved in the differentiation of palisade mesophyll cells, wound response elements, and cell cycle regulation elements, each of which occurs only once. At the same time, a large number of CREs potentially involved in hormonal responses were identified in several *PdGARP* promoters, including those responsive to ABA, MeJA, gibberellin (GA), indole acetic acid (IAA), and salicylic acid (SA). Numerous putative CREs associated with abiotic stresses were also found in many *PdGARP* promoter regions, such as those related to low-temperature response, defense and stress responses, and drought tolerance. Furthermore, there are various cis-elements associated with plant growth and development, including those involved in zein metabolism regulation, endosperm expression, seed-specific regulation, and meristem expression.

### 3.5. Gene Expression in Response to Phosphorus and Nitrogen Deficiency

To explore the expression pattern of six *NIGT1/HRS1/HHO* genes in response to nitrogen deficiency treatment, their gene expression levels were analyzed by RT-qPCR (Figure 7). At 1 d under nitrogen deficiency, the expression of four *NIGT1/HRS1/HHO* genes (*PdGARP15*, *20*, *24*, and *52*) significantly decreased. At 4 d, the expression of five out of six *NIGT1/HRS1/HHO* genes significantly increased under nitrogen deficiency, while the transcription of *PdGARP20* was repressed. These results suggest that the *NIGT1/HRS1/HHO* genes exhibit varying levels of response to nitrogen deficiency, indicating that different members of this gene family have distinct stress tolerance profiles.

To analyze the response regulation of *PHR/PHL* genes to phosphorus deficiency, low phosphorus stress experiments were conducted. Following the treatment, the expression of *PHR/PHL* genes was assessed using RT-qPCR (Figure 8). After 1 day of low phosphorus treatment, the expression of *PHL* genes decreased. At 4 days, gene expression began to rise. After 7 days of low phosphorus treatment, the expression of several *PHR/PHL* genes (*PdGARP1*, *7*, *14*, *40*, *41*, *44*, and *55*) continued to increase, while others (*PdGARP5*, *11*, *17*, *19*, *22*, *23*, *35*, *36*, *38*, *39*, *45*, *56*, and *57*) decreased.

### 3.6. KAN Genes Showed Tissue-Specific Expression

To elucidate the expression patterns of the KAN subfamily in different tissues, RT-qPCR was used to assess the relative expression levels of *KAN* genes in different tissues (Figure 9). Consistent with predictions, the RT-qPCR results showed that *KAN* genes were more highly expressed in leaves compared to roots and stems. Therefore, the expression of *KAN* genes exhibited tissue specificity.

### 3.7. Structural Analysis and Functional Validation of GLKs

The protein sequence alignment depicts the conservation of the GLK subfamily in the *P. deltoides*, and differences were observed between each member (Appendix A). The conserved domain Weblogo analysis results show that the conserved region contains 50 amino acids. Among them, the leucine residue (L) is highly retained at position 7. In addition, threonine (T) at position 4, proline (P) at position 5, histidine (H) at position 8, amino acid (V) at position 12 and 23, and alanine (A) at position 1 are also highly conserved. The secondary structure of the GLK subfamily proteins in *P. deltoides* was analyzed, revealing that all proteins consist of α-helices, extended chains, and irregular coils (Table 2). The proportion of α-helices is 13.01% to 25.32%, with the largest proportion in *PdGARP3* and the lowest proportion in *PdGARP4*. Irregular curls range from 69.80% to 85.97%, and extended strands range from 1.02% to 5.56%. These findings are consistent with secondary structure predictions. The results suggest that the tertiary structure of GLK subfamily proteins is mainly composed of α-helices and random helices, with differences in their spatial structures, likely reflecting their distinct biological functions (Figure 10).

To specifically silence the expression of target genes, we selected the coding sequence (CDS) near the 3’ end of the target genes for RNAi interference (RNAi). We chose functional candidate genes from the GLK subfamily, including *PdGARP3*, *PdGARP4*, *PdGARP8*, *PdGARP9*, *PdGARP21*, *PdGARP30*, *PdGARP46*, and *PdGARP47*, for RNAi-based silencing. The RNAi binary vector pANDA35HK was constructed and used to generate transgenic RNAi poplar plants driven by the CaMV 35S promoter.

The transgenic RNAi line was identified using RT-qPCR, and the RNAiGLK-1 line was obtained for further experiments. RT-qPCR results indicated that the expression of the *GLK* genes in the transgenic line *RNAiGLK-1* was significantly inhibited, confirming the successful generation of a GLK inhibitory expression line (Figure 11a).

## 4. Discussion

The GARP transcription factor superfamily is a large group of plant-specific transcription factors that include a variety of proteins determining their multiple functions. It consists of five subfamilies, each with its own conserved signature motif. A total of 58 PdGRAP members were identified in *P. deltoides*. Their physicochemical characteristics, genomic locations, conserved motifs, gene structures, gene duplications, and synteny were analyzed. In addition, their *cis*-elements were predicted. A total of 1274 CREs were identified, including those associated with light responsiveness, hormonal responses, abiotic stresses, and plant growth and development. The evolutionary relationships of *GARP* genes were also analyzed. Several genes, such as *PdGARP18*, *PdGARP21*, and *PdGARP51*, which are important in the evolution of the *GARP* gene family, have been identified. Their differing collinearity shows varying degrees of importance in the evolutionary process. The GARP transcription factor superfamily plays a pivotal role in plant responses to abiotic stresses. Phosphorus and nitrogen deficiency are common abiotic stresses [44,45]. Understanding how plants respond to phosphorus and nitrogen deficiency is crucial [46]. In this study, members of the PHR1/PHL1 subfamily were involved in the response to phosphorus deficiency in poplar. RT-qPCR analysis showed that, as the treatment time increased, the expression levels of all genes initially decreased and then became unregulated. This change indicates a dynamic response of the *PHR1/PHL1* genes to phosphorus deficiency. Members of the *NIGT1/HRS1/HHO* genes were involved in the response to nitrogen deficiency in poplar. The repressed expression of *PdGARP20* may form a negative feedback loop in nitrogen stress response, which deserves further exploration. Additionally, the differential regulation of these genes highlights the need to explore the specific molecular interactions and regulatory networks that govern their expression under conditions of phosphorus and nitrogen deficiency.

The GARP transcription factor superfamily is crucial for the development of plant organs [47]. In this study, we focused on the KAN subfamily, which regulates organ polarity in leaves. The expression patterns of *KAN* genes were verified in different tissues by RT-qPCR. *KAN* genes were more highly expressed in leaves compared to roots and stems. The results showed that *KAN* genes exhibited the highest expression levels in leaves, underscoring their crucial role in leaf development processes. The differential expression of *KAN* genes in leaves suggests that each *KAN* gene may play a specialized role in regulating different aspects of leaf development, such as polarity, morphology, and venation patterns.

The oldest identified function of the GARP transcription factor superfamily is its role in chlorophyll synthesis, which is primarily dependent on the GLK subfamily. In *Arabidopsis*, overexpression of *GLK* genes enhances chlorophyll biosynthesis, while the loss of both *GLK* genes results in reduced chlorophyll content and a light green leaf phenotype. In this study, we obtained a *GLK* transgenic RNAi line and measured its chlorophyll content. Compared to the wild-type plants, the levels of total chlorophyll, chlorophyll a, chlorophyll b, and carotenoids significantly decreased in the *RNAi-1* line. These results further confirm the critical role of *GLK* genes in regulating chlorophyll and carotenoid biosynthesis. These observations provide additional evidence for the pivotal role of *GLK* genes in chloroplast development and the regulation of pigment synthesis in plants.

## 5. Conclusions

In this study, we completed a comprehensive genome-wide analysis and molecular characterization of the 58 GARP members in *P. deltoides* for the first time. In addition, RT-qPCR data showed that the NIGT1/HRS1/HHO subfamilies responds to nitrogen deficiency, while the PHR/PHL subfamilies responds to phosphorus deficiency. The findings suggest that PdGARPs may be involved in nutrient signaling and stress response pathways in *P. deltoides*. The GARP transcription factor superfamily plays a vital role in regulating *P. deltoides*’ resistance to nutrient stresses. However, additional investigations are required to confirm the functional roles of these core genes. The present results will enhance the understanding of the evolution of the GARP family genes and provide valuable candidate genes for further studies of the transcriptional regulation mechanism in response to nutrient stresses in *P. deltoides*. These findings may provide insights for improving the stress resistance of poplars and for further exploring the potential applications of these candidate genes in forestry and ecosystem management.

## Figures and Tables

**Figure 1 genes-16-00322-f001:**
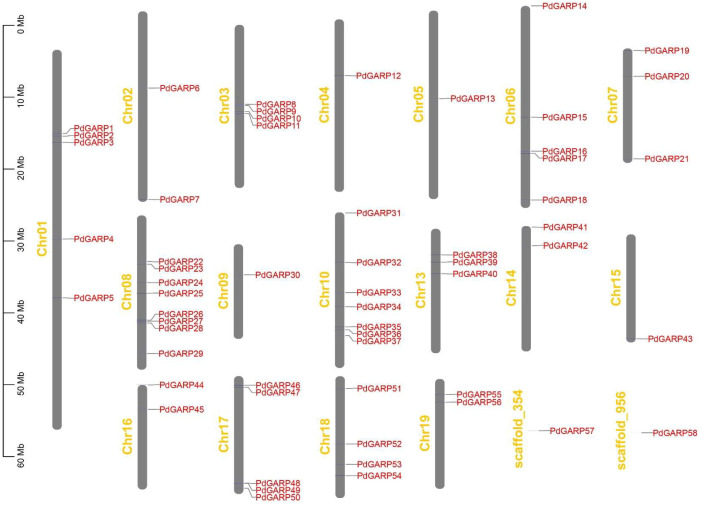
A total of 58 *PdGARP* genes are mapped onto the chromosomes. The left scale represents the chromosome lengths in megabases (Mb). “Chr” refers to chromosomes, and the number indicates the chromosome number.

**Figure 2 genes-16-00322-f002:**
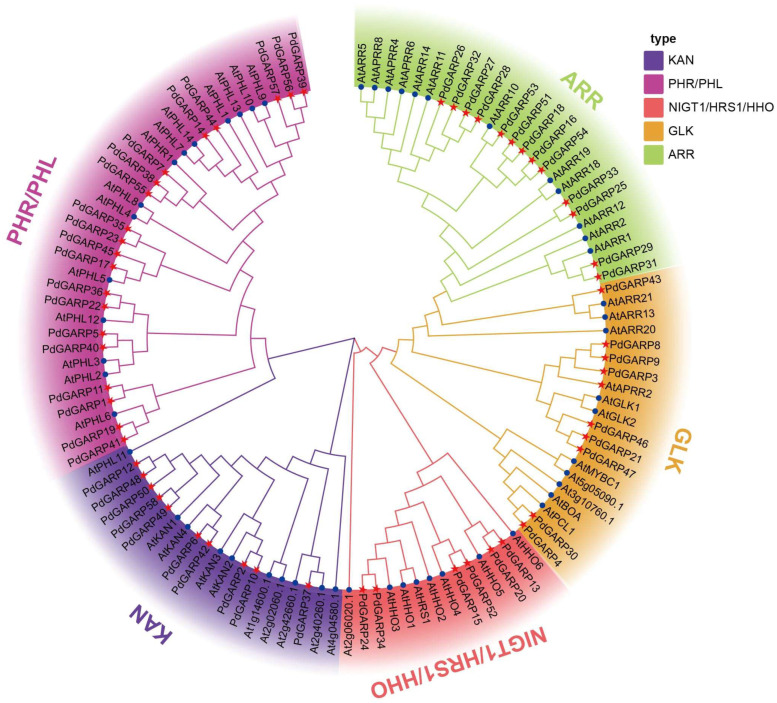
Phylogenetic tree constructed with GARP proteins from *P. deltoides* and *A. thaliana* by neighbor-joining method. The phylogenetic tree includes the PHL, KAN, HHO, GLK, and ARR subfamilies, with background colors representing each subfamily: violet, indigo, red, orange, and green. The blue circle indicates *A. thaliana*, while the red star indicates *P. deltoides*.

**Figure 3 genes-16-00322-f003:**
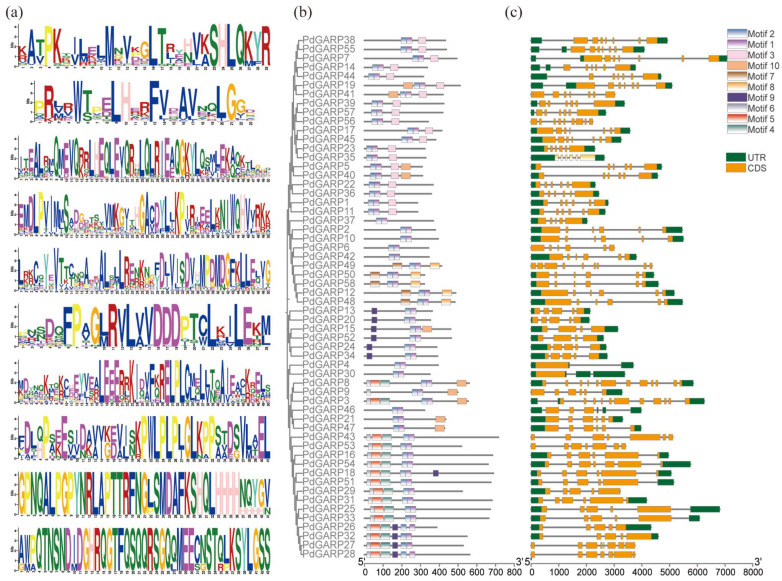
Gene structure and conserved motifs of GARP proteins in *P. deltoides*. (**a**) Sequence characterization and molecular identification of PdGARPs. The height of the letter indicating amino acid residue at every position reflects the degree of conservation. The numbers along the vertical axis indicate the positions of residue within the motifs. Meanwhile, the horizontal axis represents the content quantified in bits. (**b**) Motif composition of PdGARP proteins, with each motif highlighted in different colored boxes. (**c**) Exon–intron organizations of PdGARPs, with different regions indicated by color codes: green for untranslated regions, orange for exons, and black lines for introns. The scale at the bottom represents the gene length in kilobases (kb).

**Figure 4 genes-16-00322-f004:**
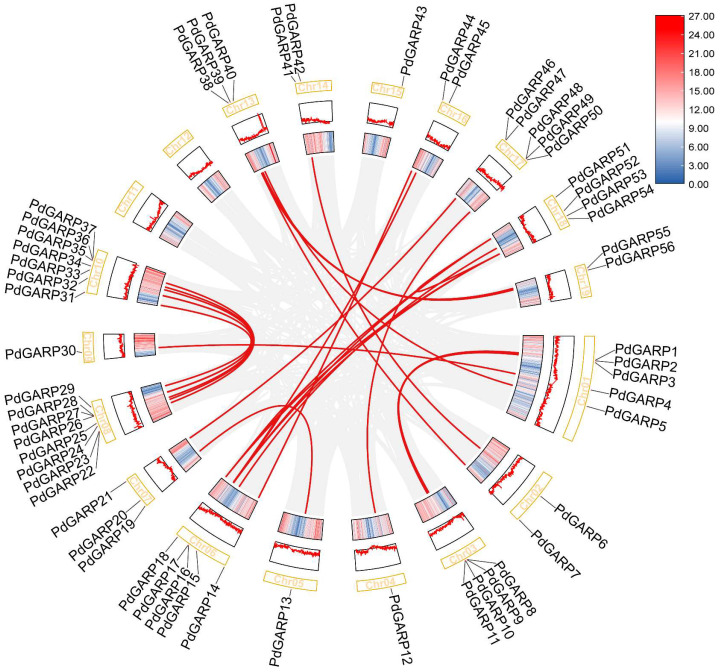
Collinearity analysis of PdGARPs. The gray curves indicate the collinear regions, while the red curves highlight the gene pairs that have undergone segmental duplication.

**Figure 5 genes-16-00322-f005:**
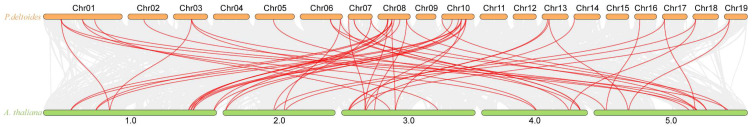
Collinearity analysis between the *P. deltoides* and *A. thaliana* genomes. The gray lines in the background represent the collinearity blocks between species, while the collinear gene pairs with *GARP* genes are highlighted in the red lines. The sandy-brown and olivine rectangles represent the chromosomes of *P. deltoides* and *A. thaliana*, respectively.

**Figure 6 genes-16-00322-f006:**
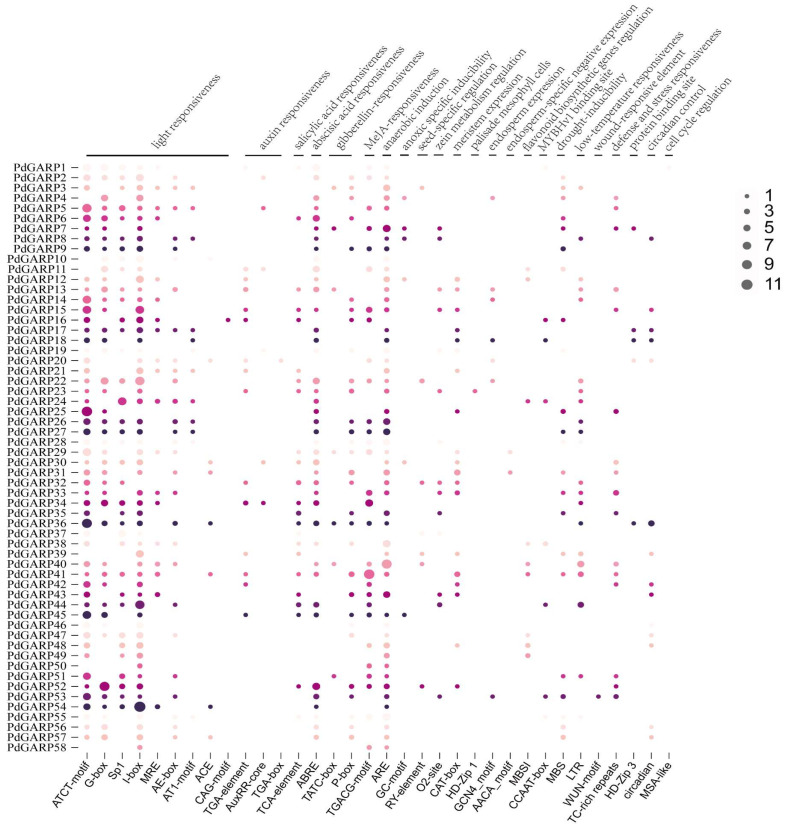
Distribution of *cis*-elements in the promoter regions of *PdGARPs*. Different sized and colored circles represent *cis*-element numbers.

**Figure 7 genes-16-00322-f007:**
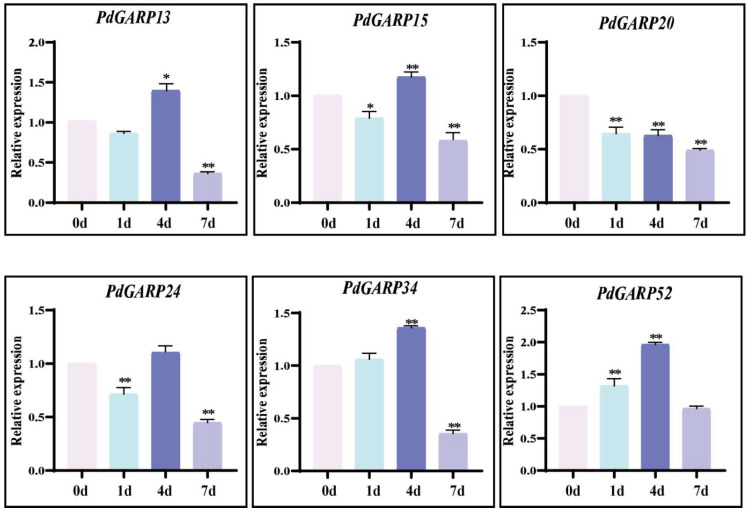
Expression analysis of *NIGT1/HRS1/HHO* genes under nitrogen deficiency. The relative expression levels of the *NIGT1/HRS1/HHO* genes at 0, 1, 4, and 7 days were measured using RT-qPCR. Asterisks indicate statistically significant differences: *, *p*-value ≤ 0.05; **, *p*-value ≤ 0.01, determined by Student’s *t*-test. Values are presented as means ± standard deviation.

**Figure 8 genes-16-00322-f008:**
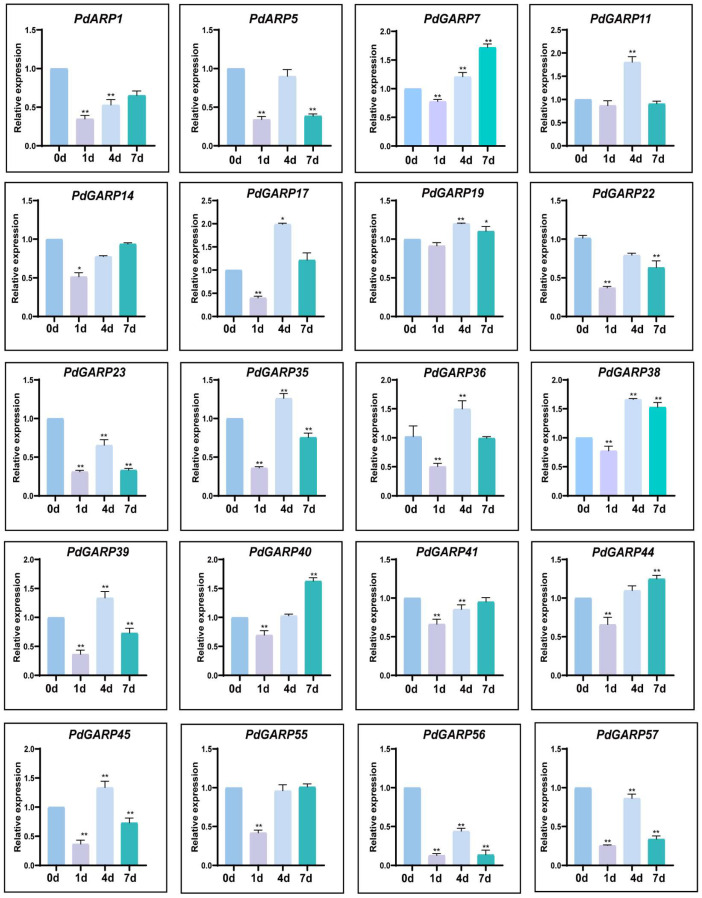
Expression patterns of *PHR/PHL* genes in *P. deltoides* under phosphorus deficiency treatment. The relative expression levels of *PHR/PHL* genes were analyzed using RT-qPCR. Asterisks indicate statistically significant differences: *, *p*-value ≤ 0.05; **, *p*-value ≤ 0.01, determined by Student’s *t*-test. Values are presented as means ± standard deviation.

**Figure 9 genes-16-00322-f009:**
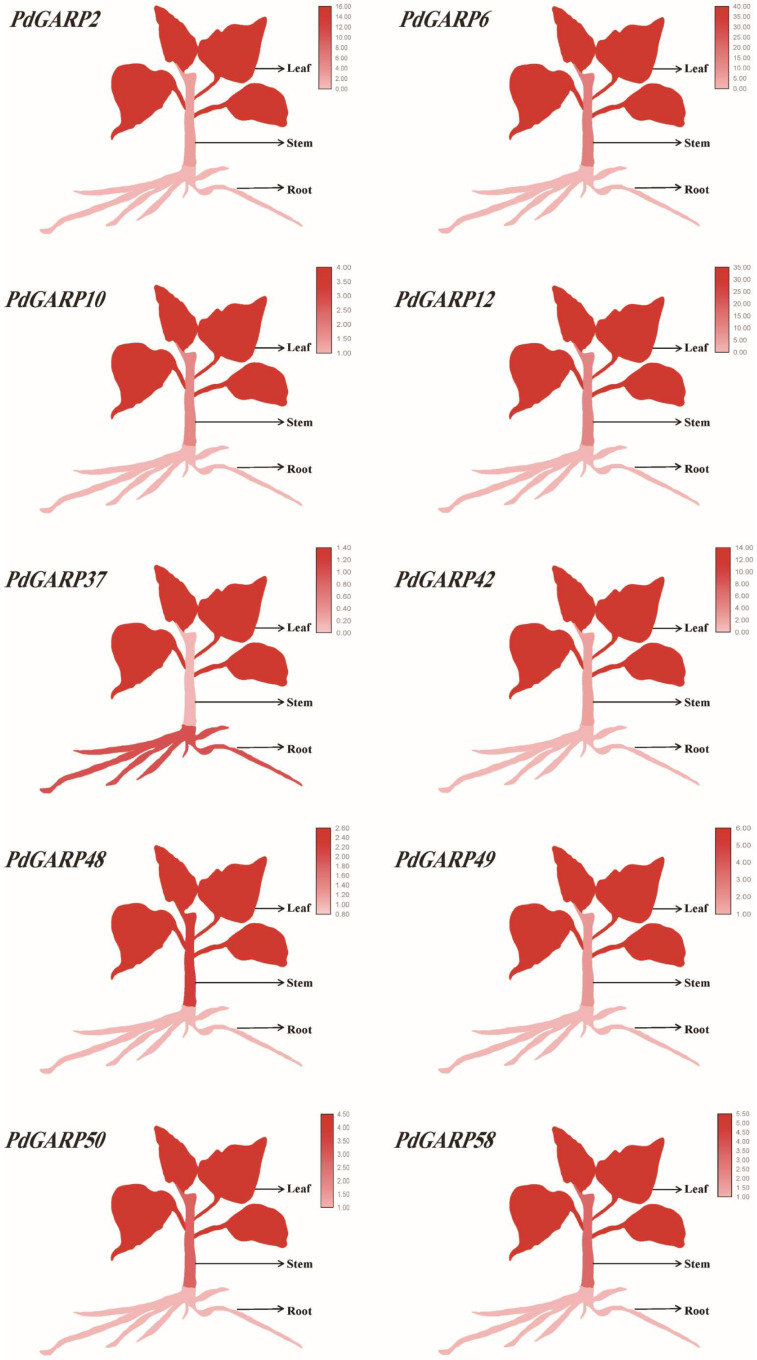
Gene expression of *KAN* genes in root, stem, and leaf of *P. deltoides*. The gene expression levels were assessed using RT-qPCR data. The heatmap shows gene expression patterns, with dark red indicating high expression and light pink indicating low expression.

**Figure 10 genes-16-00322-f010:**
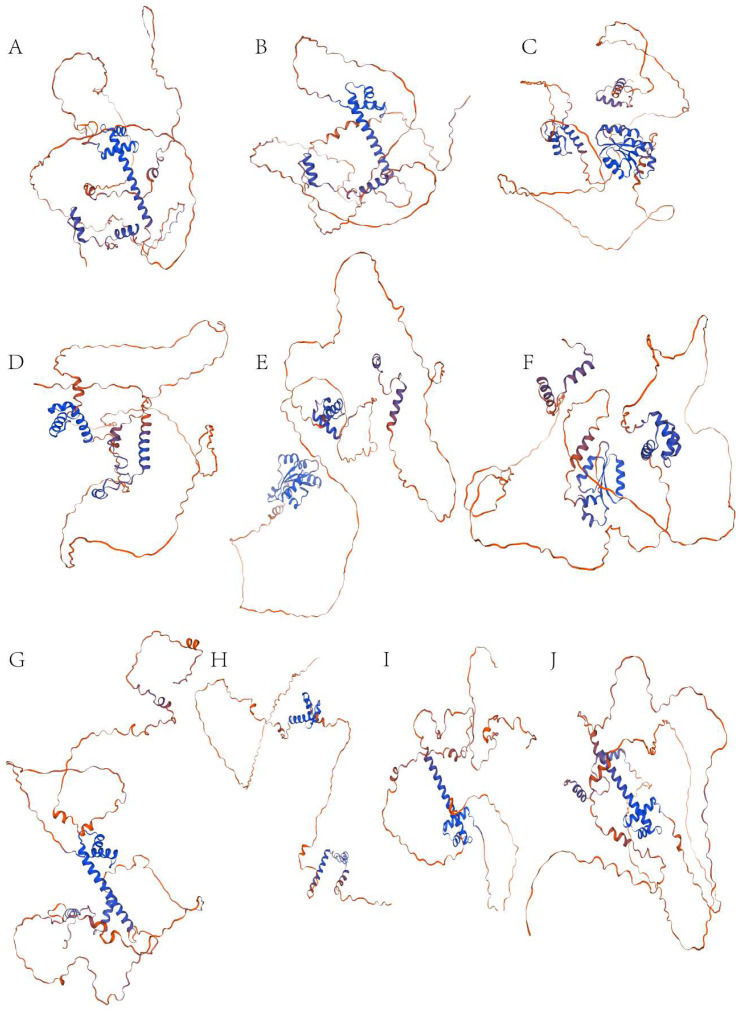
Predicted tertiary structure of AtGLK1, AtGLK2 and GLK subfamily members. (**A**–**J**) represent the proteins AtGLK1, AtGLK2, PdGARP3, PdGARP4, PdGARP8, PdGARP9, PdGARP21, PdGARP30, PdGARP46, and PdGARP47, respectively.

**Figure 11 genes-16-00322-f011:**
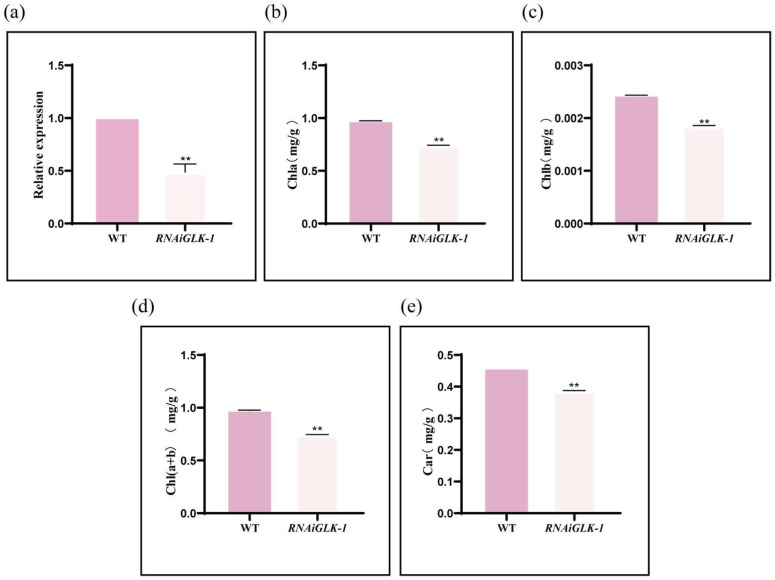
Relative expression and pigment contents in WT and transgenic lines. *RNAiGLK-1*, repression line of *PdGLKs*; WT, wild type. Asterisks indicate statistically significant differences: **, *p*-value ≤ 0.01, determined by Student’s *t*-test. Values are presented as means ± standard deviation. (**a**) The relative gene expression levels of WT and *RNAiGLK-1*. (**b**–**e**) The contents of chlorophyll a, chlorophyll b, carotenoid, and total chlorophyll in the WT and transgenic lines, respectively. To investigate the changes in pigment content of GLKs in *P. deltoides*, we measured the contents of total chlorophyll, chlorophyll a, chlorophyll b, and carotenoids. Compared to the wild-type (WT) plants, the levels of total chlorophyll, chlorophyll a, chlorophyll b, and carotenoids significantly decreased in the RNAi-1 lines (**b**–**e**).

**Table 1 genes-16-00322-t001:** Ka/Ks analysis for the *PdGARP* genes identified in *P. deltoides*. SD, mean segmental duplication; TD, mean tandem duplication. Ka/Ks ratio > 1, Ka/Ks = 1, and Ka/Ks < 1 indicate positive selection, neutral evolution, and purifying selection, respectively.

Duplicated Pairs	Ka	Ks	Ka/Ks	Duplication Type	Purifying Selection
PdGARP1–PdGARP11	0.0654	0.2315	0.2824	SD	YES
PdGARP2–PdGARP10	0.0889	0.3094	0.2874	SD	YES
PdGARP3–PdGARP8	0.1031	0.266	0.3875	SD	YES
PdGARP4–PdGARP30	0.082	0.3164	0.2592	SD	YES
PdGARP5–PdGARP40	0.0329	0.1629	0.2019	SD	YES
PdGARP7–PdGARP38	0.3542	1.4764	0.2399	SD	YES
PdGARP6–PdGARP42	0.1022	0.1713	0.5966	SD	YES
PdGARP12–PdGARP48	0.0577	0.2624	0.22	SD	YES
PdGARP13–PdGARP20	0.0817	0.2609	0.3133	SD	YES
PdGARP16–PdGARP18	0.3795	1.3808	0.2748	TD	YES
PdGARP14–PdGARP44	0.0462	0.1789	0.2585	SD	YES
PdGARP16–PdGARP54	0.0888	0.2349	0.378	SD	YES
PdGARP16–PdGARP51	0.3853	1.4246	0.2705	SD	YES
PdGARP15–PdGARP52	0.0617	0.3063	0.2014	SD	YES
PdGARP18–PdGARP51	0.0623	0.2367	0.2632	SD	YES
PdGARP18–PdGARP54	0.374	1.4315	0.2612	SD	YES
PdGARP21–PdGARP46	0.0879	0.3627	0.2423	SD	YES
PdGARP29–PdGARP31	0.0513	0.2653	0.1934	SD	YES
PdGARP23–PdGARP35	0.0614	0.1651	0.3717	SD	YES
PdGARP27–PdGARP32	0.1228	0.3771	0.3257	SD	YES
PdGARP24–PdGARP34	0.0678	0.264	0.257	SD	YES
PdGARP25–PdGARP33	0.0618	0.2468	0.2503	SD	YES
PdGARP26–PdGARP32	0.0673	0.3397	0.1982	SD	YES
PdGARP38–PdGARP55	0.098	0.1965	0.499	SD	YES
PdGARP39–PdGARP56	0.0615	0.3063	0.2009	SD	YES
PdGARP51–PdGARP54	0.382	1.3708	0.2787	TD	YES

**Table 2 genes-16-00322-t002:** Predicted secondary structure of GLK subfamily members.

Gene Name	α Helix	Extended Strand	Random Coil
AtGLK1	20.18%	3.67%	76.15%
AtGLK2	19.43%	1.81%	78.76%
PdGARP3	25.32%	4.88%	69.80%
PdGARP4	13.01%	1.02%	85.97%
PdGARP8	24.01%	5.56%	70.43%
PdGARP9	21.69%	5.42%	72.89%
PdGARP21	19.40%	3.23%	77.37%
PdGARP30	15.47%	2.29%	82.23%
PdGARP46	21.56%	4.69%	73.75%
PdGARP47	21.78%	2.81%	75.41%

## Data Availability

The original contributions presented in the study are included in the article/Appendix A; further inquiries can be directed to the corresponding author.

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
