# Peer review of "Genome-Wide Identification and Comprehensive Analysis of the GARP Transcription Factor Superfamily in Populus deltoides"

_genes, 2025, doi:10.3390/genes16030322_

Round 1
Reviewer 1 Report
Comments and Suggestions for Authors
The GARP transcription factor superfamily, a plant-specific category, was found in Populus deltoides by bioinformatics techniques. Fifty-eight PdGARP proteins were categorized into five subfamilies according on evolutionary relationships and protein sequence similarity. Segmental duplication was identified as the main catalyst for the family's proliferation. Thirty-two gene pairs were common between P. deltoides and Arabidopsis thaliana. An extensive examination of cis-regulatory elements (CREs) in PdGARP promoters identified 1,274 CREs, including those sensitive to light, anaerobic conditions, ABA, and MeJA. The differential expression patterns of NIGT1/HRS1/HHO and PHR/PHL subfamily members indicate their participation in stress reactions revealing that the NIGT1/HRS1/HHO subfamilies respond to nitrogen deficiency, while the PHR/PHL subfamilies respond to phosphorus deficiency. Further research is needed to confirm these roles.
Lines 291, 328: Italic: P. deltoides
Line 312: tertiaryy
Author Response
Comments 1: [Lines 291, 328: Italic: P. deltoides
Line 312: tertiaryy.]
We sincerely thank you for careful reading and pointing this out. We agree with this comment. Therefore, we have fixed a few of these formatting errors. We have corrected the “ P. deltoides” into“ P. deltoides” in line 291 and 328, the “ tertiaryy” into “ tertiary” in line 312. This makes our manuscript more standardized and scientific. Again, thank you very much for your careful review.
Reviewer 2 Report
Comments and Suggestions for Authors
The paper is interesting and represents a considerable amount of novel data for a woody plant. The bioinformatics approach is very complete and provides valuable information, although the wet experiments could be improved.
Major points:
- Which genes were used as references in the qRT-PCR experiments? Please include this information in the methods section.
- Which statistical analyses have been performed in figures 7, 8, and 12? What do the asterisks mean? What do the asterisks mean in figures 8 and 12?
- Figure 12 is the functional validation by constructing a transgenic line; the problem is that there is only a single line, and usually, data from at least three independent transgenic lines are required. In addition, there is no information on how the lines have been constructed in the methods section. Authors should present the complete data on transgenic plants, but there is enough data in the paper, so I suggest removing the figure.
- Figures 1, 3, 4, 5, and 6 are very small; please enlarge them so the lettering can be read.
- Line 277: phosphorous deficiency.
Reviewer 3 Report
Comments and Suggestions for Authors
The work is aimed to perform a comprehensive analysis of the GARP transcription factor superfamily in Populus deltoides.
The methodological approach is consistent with the research aim.
Results suggest that PdGARPs may be involved in nutrient signaling and stress response pathways in P. deltoides.
References are appropriate.
However, the following revisions are suggested:
- In the “Introduction” section,
- Some recent works, such as that about the GLK transcription factors available at https://doi.org/10.3390/d14030228 could be included
- In the “Results section”
- Figures 3 is not readable. I suggest to include them as supplementary material or to find a great solution.
- Figure 10 could be included as supplementary material.
- In the “Discussion” section,
- The sentence at lines 341-342 is a conclusive sentence. I suggest to move it in the next section.
- Furthermore, it is necessary to better underline the value added of the research compared to previous works.
- In the “Conclusion” section,
- Please highlight the impact and the benefits of the achieved results.
Overall, it is necessary a better organization of the sentences.
Some minor issues:
- Please check the English form. There are short sentences that are often unconnected to each other.
Comments on the Quality of English Language
There are short sentences that are often unconnected to each other.
Author Response
Thanks for your careful checks. Based on your comments, we have made the corrections.
Comments 1: [In the “Introduction” section, Some recent works, such as that about the GLK transcription factors available at https://doi.org/10.3390/d14030228 could be included.]
We totally agree with this comment. In the revised manuscript, it has been added as the references as seen in page 2. This literature complements the functionality of GLK genes and makes my citations more sufficient.
Comments 2: [In the “Results section” Figures 3 is not readable. I suggest to include them as supplementary material or to find a great solution. Figure 10 could be included as supplementary material.]
Thanks for the suggestion. We have made adjustments to the font size, color combination, and sharpness of figure 3. We hope to put it in the manuscript. We have put Figure 10 into the supplementary material.
Comments 3: [In the “Discussion” section, The sentence at lines 341-342 is a conclusive sentence. I suggest to move it in the next section. Furthermore, it is necessary to better underline the value added of the research compared to previous works.]
We thought it was a good suggestion and made a change. We added a few sentences to emphasize the added value of our research. This section is on page 16 of the resubmitted manuscript.
Comments 4: [In the “Conclusion” section, Please highlight the impact and the benefits of the achieved results.]
As you are concerned, we have revised this section to clearly highlight our contributions and the impact of the achieved results. This section is on page 17 of the resubmitted manuscript.
Thank you again for this valuable comment.